# TaTLP1 interacts with TaPR1 to contribute to wheat defense responses to leaf rust fungus

Fei Wang[1,☉], Shitao Yuan[1,☉], Wenyue Wu[1,☉], Yiqing Yang[1], Zhongchi Cui[1], Haiyan Wang[1]*, Daqun Liu[1,2]*

**1** Center of Plant Disease and Plant Pests of Hebei Province, College of Plant Protection, Hebei Agricultural University, Baoding, China, **2** Graduate School of Chinese Academy of Agricultural Sciences, Beijing, China

☉ These authors contributed equally to this work.
* ndwanghaiyan@163.com (HW); 1468135313@qq.com (DL)

**Data Availability Statement:** All relevant data are within the manuscript and its Supporting Information files.

**Funding:** This work was funded by the National Natural Science Foundation of China (No.

## Abstract

Thaumatin-like proteins (TLPs), which are defined as pathogenesis-related protein family 5 (PR5) members, are common plant proteins involved in defense responses and confer antifungal activity against many plant pathogens. Our earlier studies have reported that the *TaTLP1* gene was isolated from wheat and proved to be involved in wheat defense in response to leaf rust attack. The present study aims to identify the interacting proteins of TaTLP1 and characterize the role of the interaction between wheat and *Puccinia triticina* (*Pt*). Pull-down experiments designed to isolate the molecular target of TaTLP1 in tobacco resulted in the identification of TaPR1, a pathogenesis-related protein of family 1, and the interaction between TaTLP1 and TaPR1 was confirmed by yeast two-hybrid experiments (Y2H), bimolecular fluorescence complementation (BiFC), and co-immunoprecipitation (Co-IP). *In vitro*, TaTLP1 and TaPR1 together increased antifungal activity against *Pt*. *In vivo*, the disease resistance phenotype, histological observations of fungal growth and host responses, and accumulation of $H_2O_2$ in TaTLP1-TaPR1 in co-silenced plants indicated that co-silencing significantly enhanced wheat susceptibility compared to single knockdown TaTLP1 or TaPR1 plants. The accumulation of reactive oxygen species (ROS) was significantly reduced in co-silenced plants compared to controls during *Pt* infection, which suggested that the TaTLP1-TaPR1 interaction positively modulates wheat resistance to *Pt* in an ROS-dependent manner. Our findings provide new insights for understanding the roles of two different PRs, TaTLP1 and TaPR1, in wheat resistance to leaf rust.

## Author summary

In host-pathogen interactions, novel proteins that are specifically involved in pathological response are collectively referred to as "pathogenesis-related proteins" (PRs). PR proteins encoded by plant defense genes play key roles in plant defense responses, particularly in systemic-acquired resistance. However, their molecular roles in the response of wheat to fungal infection are typically not well characterized. PR5 family proteins are referred to as thaumatin-like proteins (TLPs), are able to rapidly accumulate to high levels in response to biotic or abiotic stress and exhibit antifungal activity in various plant species. Thus,

31501623) (http://www.nsfc.gov.cn/), the corresponding author, Haiyan Wang, received the funding. The funder had no role in study design, data collection and analysis, or preparation of the manuscript.

**Competing interests:** The authors have declared that no competing interests exist.

investigation of PR5 proteins is crucial for understanding the mechanism of wheat defense against *Pt* and searching for new ways to control the wheat leaf rust disease.

## Introduction

Plenty of studies have demonstrated that pathogenesis-related (PR) proteins play key roles in plant disease-resistance responses and specifically systemic-acquired resistance (SAR) [1]. Half a century ago, PRs were found to play a pioneering role in the discovery of plant innate immunity as different gene families were discovered to be responsible for the accumulation of PRs upon pathogen challenge. PRs and similar proteins have been reported to be induced by various types of pathogens in many plants. So far, a total of 17 gene families designated as PR genes were identified from plant species in response to various pathogens [2–4]. Among different groups of PRs, PR1, PR2, and PR5 are the most common and accumulate locally and systemically suggesting an involvement in SAR that represents a form of plant immunization [2]. *PRs* generally encode small antimicrobial proteins and their expression levels increase quickly when stimulated by biotic or abiotic stress, and therefore serve as a marker of plant immune signalling. However, the biochemical activity and mode of action of PR proteins has remained elusive.

PR5 family proteins, which are also referred to as thaumatin-like proteins (TLPs), are able to rapidly accumulate to high levels in response to biotic or abiotic stress and confer antifungal activity in various plant species [5–6]. Most TLPs respond to some biotic or abiotic stresses and some resistance genes function together with TLPs based on genome-wide and functional network analyses in *Arabidopsis*, *Oryza*, *Populus*, *Zea*, *Physcomitrella* and *Chlamydomonas* [7–8]. The PR5 family protein osmotin has been identified as an interacting partner for BcIEB1, a glycoprotein abundantly secreted by *Botrytis cinerea*, and this interaction protects *B.cinerea* against the antifungal action of osmotin and also modulates the elicitor activity of BcIEB1 in plants [9]. GhPR5, a partner protein of PevD1 which is an Alt a 1-like protein, was isolated from cotton (*Gossypium hirsutum*) plants by co-purification assays, and the PevD1-GhPR5 interaction was determined to occur at the C-terminus of PevD1 by pull-down and yeast two-hybrid techniques [10]. A new osmotin-like protein (OLP) gene of the PR5 family from *Solanum nigrum*, *SindOLP*, was cloned and exhibited *in vitro* antifungal activity against *Macrophomina phaseolina* [11]. Transgenic sesame lines overexpressing *SindOLP* showed enhanced resistance against drought and salinity stresses and biotic stress caused by a fungal pathogen [12].

PR1 proteins represent a large protein family (23 in wheat) that are upregulated early in the defense response. Findings from a recent study provided genetic and biochemical evidence that indicates that PR1 binds to sterols and inhibits pathogen growth [13]. A dimeric PR1 type protein (PR-1-5) was found to interact with *Stagonospora nodorum* ToxA and potentially mediate ToxA-induced necrosis in wheat. Subsequent mutational analysis identified several residues on both ToxA and TaPR-1-5 that are required for their interaction as well as the induction of necrosis, which showed that SnToxA interacts specifically with TaPR-1-5 [14]. Interestingly, analysis of the SnTox3-TaPR-1 interaction demonstrated that SnTox3 can interact with a broad range of TaPR-1 proteins [15]. Recently, PR1 was found to interact with SsCP1, a Cerato-platanin protein (CP), in the apoplast to facilitate infection by the necrotrophic phytopathogen *Sclerotinia sclerotiorum* using various methods including yeast two-hybrid, Glutathione S-transferase (GST) pull-down, co-immunoprecipitation (CoIP), and bimolecular florescence complementation (BiFC). Overexpressing *PR1* enhanced resistance to

the wild-type strain, but not to the *Sscp1* knockout strain of *S. sclerotiorum* [16]. Collectively, the finding that both ToxA and SsCP1 interact with PR1 proteins is suggestive of the critical nature of these interactions and implies that PR1 proteins play a key role in mediating disease outcomes in the plant-fungus interaction.

Wheat leaf rust, caused by the obligate biotrophic plant pathogen *Puccinia triticina* (*Pt*), occurs world-wide and is one of the most destructive diseases of wheat in major wheat-growing regions [17]. Wheat leaf rust can reduce the quality of harvested grain and cause significant yield losses. Thus, it is crucial to understand the defense mechanisms of wheat against *Pt* and to search for new ways to control wheat leaf rust disease. Resistance of wheat cultivars against *Pt* is usually divided into two genetic categories, seedling resistance genes and adult plant resistance (APR) genes also referred to as minor genes. Seedling resistance genes are referred to as major resistance genes and they confer resistance at both the seedling and adult plant stages. Major resistance genes are critical for expression of a hypersensitive reaction (HR), and are often race-specific and mediated by gene-for-gene resistance [18–19]. APR genes are commonly detected at post-seedling stages, are quantitatively inherited with partial effects on disease severity (DS), and are not involved in express of a HR [20–21]. As we know, regardless of seedling resistance genes or APR genes, our findings strongly suggest that PRs are involved in wheat rust defense responses [22–24].

However, the mechanistic understanding of how rust can interact directly with host proteins remains limited because of the large genome size, polyploid nature, and high content of repetitive DNA of hexaploid wheat (*T. aestivum* L.) obfuscating functional gene characterization efforts. Recently Yang et al. reported that an effector, Pst18363, was identified from *Puccinia striiformis* f. sp. *tritici* (*Pst*) and shown to interact with and stabilize wheat Nudix hydrolase 23, which suppresses reactive oxygen species (ROS) accumulation to facilitate *Pst* infection [25]. Huai et al. found that *Pst* stimulates ABA biosynthesis in host cells and thereby upregulates TaSTP6 (a sugar transporter in wheat) expression, which increases sugar supply and promotes fungal infection [26]. In our previous study, a thaumatin-like protein gene, *TaTLP1*, was cloned from wheat in response to the infection of *Pt* race PHNT (GenBank accession number KJ764822). TaTLP1 contained a functional secretion peptide and was secreted into the apoplast. BSMV-induced gene silencing and pathology tests on the silenced plants revealed that *TaTLP1* is involved in wheat resistance to the wheat leaf rust pathogen [24]. In this study, we aim to screen for interacting proteins of TaTLP1 using pull-down experiments and to characterize the role of the protein interactions between wheat and *Pt*. Our findings provide new insights for understanding the roles of different PRs, especially TaTLP1 and TaPR1, in wheat defense to *Pt* infection.

## Results

### Identifying interacting proteins of TaTLP1 by GFP-Trap and mass spectrometry (MS)

The ChromoTek GFP-Trap tool is based on a GFP-binding protein derived from an alpaca single variable domain antibody. It has been demonstrated to be an excellent tool to for purification of GFP-fused proteins and their interacting partners [27–28]. To identify interaction protein(s) targeted by TaTLP1, pull-down assays using the GFP-Trap kit were carried out by mixing *Agrobacterium tumefaciens* GV3101 containing a construct encoding recombinant TaTLP1-pEarlyGate103 (c-GFP tagged) with *Nicotiana benthamiana* leaf extracts, followed by purification of proteins with anti-GFP gel beads. After elution, purified proteins were separated by SDS-PAGE. Several different bands were observed between samples from TaTLP1-pEarlyGate103 and controls (Fig 1). A total of 165 proteins were identified by mass

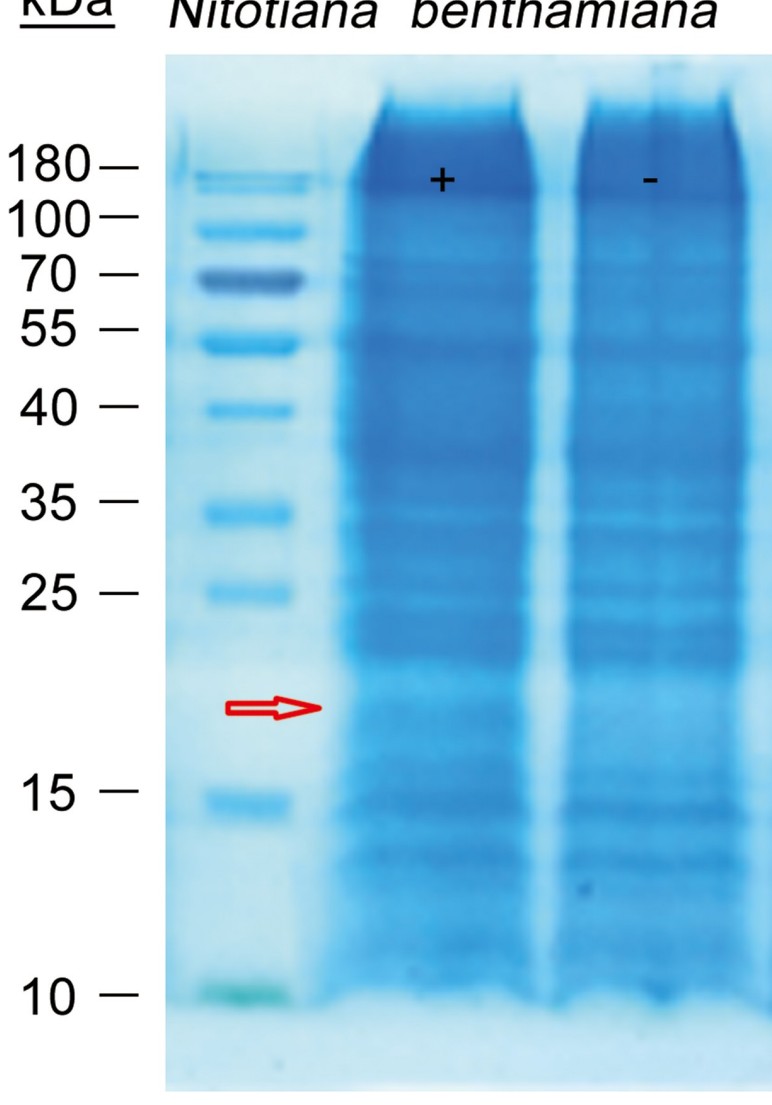

**Fig 1. GFP-Trap immunoprecipitates of TaTLP1-pEarlyGate103 with its associated proteins.** SDS-PAGE by Coomassie Brilliant Blue-staining of the proteins resulting from pull-down assays with *N. benthamiana* leaf extracts, incubated with (+) or without (-) TaTLP1. Pathogenesis-related protein 1 (PR1) co-purifies with TaTLP1, the red arrow showing 18 kD.

spectrometry in differential bands from plant extraction with or without TaTLP1. S1 Table summarizes all 7 candidates uniquely found in the TaTLP1-pEarlyGate103 expressing samples according to their biological function which have at least 15 different peptides detected in MS (S1 Table). This result provided us great confidence in the reliability of identifying the interacting proteins of TaTLP1 via GFP-Trap and MS. Among the identified proteins, a PR1-like protein was co-purified with TaTLP1-pEarlyGate103 as an 18-kD protein band was co-purified with TaTLP1, whereas no additional band appeared in the control assays with extracts alone (Fig 1). Since the expression profiles and functional analysis of *TaPR1* were performed in our previous study [29], these results inspired our hypothesis that TaTLP1 and TaPR1 are likely to interact with each other and are associated with resistance of wheat against *Pt* infection.

### *TaTLP1* and *TaPR1* are induced by leaf rust and share the same expression profiles

To further examine whether the identified proteins functionally interact with and regulate TaTLP1, TaPR1 was selected for further biochemical experiments on the basis of it being identified in all three independent mass spectrometry results. First, we quantified the relative expression levels of both defense-related genes (*TaTLP1* and *TaPR1*) in response to leaf rust infection using Thatcher which is a susceptible material to *Pt* as control. In the compatible interaction (Thatcher response to *Pt*), *TaTLP1* and *TaPR1* did not show significant changes at either time point after inoculation with *Pt* race 07-10-426-1, named as PHNT according to the international code [30]. In contrast, in the incompatible interaction (TcLr19 response to *Pt*), the transcript levels both of *TaTLP1* and *TaPR1* increased to their highest levels from 72 hours post inoculation (hpi) to 168 hpi. The transcript levels of *TaTLP1* and *TaPR1* were much higher during the incompatible interaction compared to the compatible interaction at 0–168 hpi. Based on these observations, the expression levels of *TaTLP1* and *TaPR1* in TcLr19 plants after inoculation by leaf rust were similar throughout the time course (Fig 2), suggesting that the expression of *TaTLP1* and *TaPR1* are similarly induced by leaf rust.

### TaTLP1 physically interacts with TaPR1 *in vitro*

To further assess whether TaPR1 could directly interact with TaTLP1, we conducted *in vitro* yeast-two hybrid (Y2H) assays using TaPR1 without a signal peptide as bait fused to the GAL4 DNA-binding domain (BD) and TaTLP1 without a signal peptide as prey fused to the GAL4-activation domain (AD). AvrLm1 and BnMPK9 served as a positive control with serially

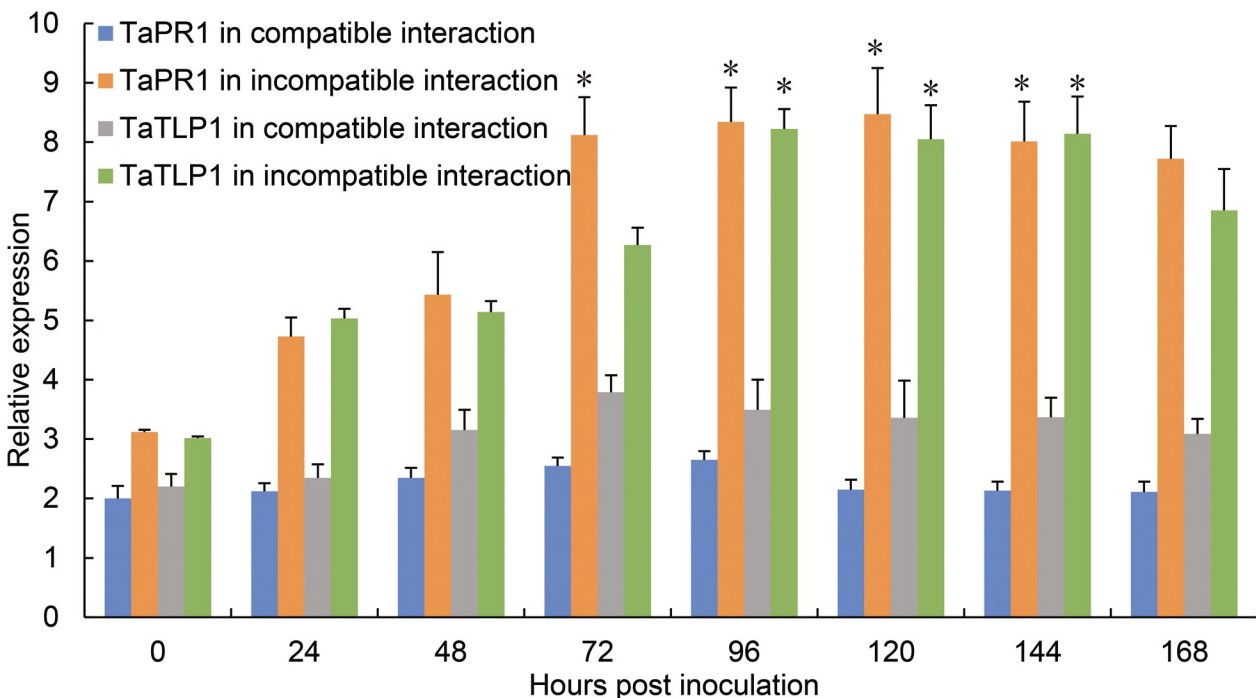

**Fig 2. Expression profiles of *TaPR1* and *TaTLP1* in incompatible and compatible interaction leaves infected by the avirulent *Pt* isolate PHNT at different times post inoculation.** The relative expression is expressed as fold change relative to mock inoculated plants at 0 hpi. The y-axis indicates the amounts of *TaPR1* and *TaTLP1* transcript normalized to the *GAPDH* gene. The x-axis indicates sampling times. The transcript levels of *TaPR1* and *TaTLP1* in compatible interaction at different times were standardized as 1. Data are means ± standard errors (SE) of three independent experiments. Differences between time-course points were assessed using Student's *t*-tests (* $p < 0.05$).

diluted cell suspensions that were used to determine the strength of the interaction by blue coloration derived from α-galactosidase activity on selective agar plates. Similar to positive controls (Fig 3, rows 1), the yeast transformant co-expressing the full-length mature TaTLP1 and TaPR1 proteins produced blue coloration (Fig 3, rows 5). In contrast, replacement of either the TaTLP1 prey or the TaPR1 bait with the yeast proteins AD or BD from empty vectors caused a loss of blue coloration (rows 2–4). These results support the hypothesis that TaTLP1 and TaPR1 are likely to interact with each other.

## Interaction between TaTLP1 and TaPR1 was confirmed by BiFC, co-localization and Co-IP assays

To validate the Y2H interaction between TaTLP1 and TaPR1, we generated BiFC constructs, ΔspTaTLP1-pENTR and ΔspTaPR1-pENTR were recombined into the binary vectors pDEST-VYNE-GW and pDEST-VYCE-GW [31]. The BiFC assay was performed by transiently co-expressing ΔspTaTLP1-VYNE-GW and ΔspTaPR1-VYCE-GW in *N. benthamiana* leaves. As shown in Fig 4A, similar to positive controls, a strong fluorescent signal was observed in cells co-expressing ΔspTaTLP1-VYNE-GW and ΔspTaPR1-VYCE-GW, but no fluorescence in the negative controls using empty vector constructs ΔspTaTLP1-VYNE-GW and ΔspTaTLP1-VYCE-GW (Fig 4A).

Then this interaction was further studied *in vivo* using co-localization techniques. Overlapping fluorescence was detected in the apoplast of agroinfiltrated cells when TaTLP1-pGWB454 and TaPR1-pEarlyGate103 were co-expressed (Fig 4B). These results indicate that TaTLP1 interacts with TaPR1 in the plant cell apoplast. Co-immunoprecipitation assays were performed upon transient expression in *N. benthaminana* to further validate the interaction between TaTLP1 and TaPR1 *in vivo*. ΔspTaTLP1-pGWB414 co-immunoprecipitated with ΔspTaPR1-pEarlyGate103, and vice versa (Fig 5A). Taken together, these observations support that TaPR1 and TaTLP1 interact *in vivo*.

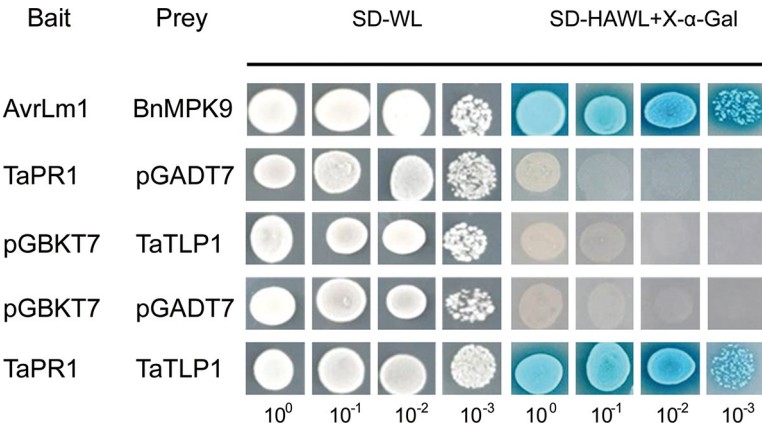

**Fig 3. Yeast two-hybrid analysis of the interaction between TaPR1 and TaTLP1.** Yeasts expressing the indicated combinations of bait and prey were spotted on the synthetic dropout medium without leucine and tryptophan (SD-WL) and SD medium without leucine, tryptophan, histidine, and adenine supplemented with X-α-Gal and Aureobasidin A (SD-HAWL + X-α-Gal). Only those co-expressing TaPR1 and TaTLP1 grew on SD-HAWL plates and developed blue coloration, indicative of an interaction between both proteins. AvrLm1 (bait) and BnMPK9 (prey) are the positive controls; TaPR1 and pGADT7, TaTLP1 and pGBKT7, pGADT7 and pGBKT7 are used as the negative controls.

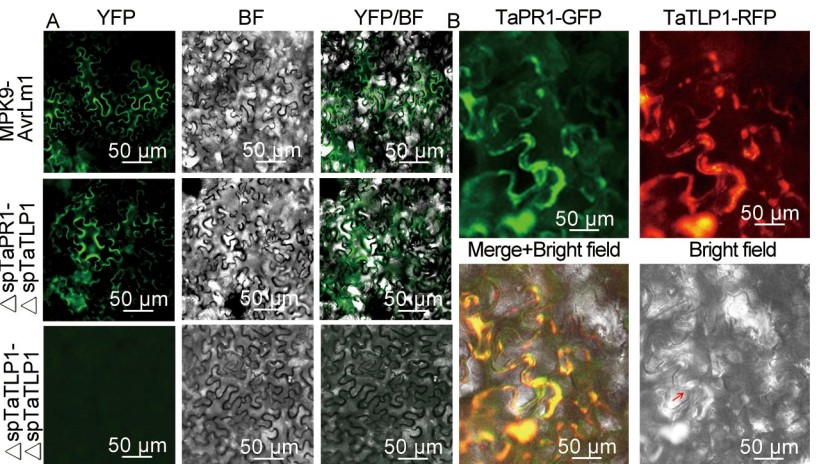

**Fig 4. TaTLP1 associates with TaPR1. (A)** BiFC confirmed the interaction between TaPR1 and TaTLP1. ΔspTaPR1-VYCE-GW and ΔspTaTLP1-VYNE-GW were transiently co-expressed in *N. benthamiana*. Co-infiltration of ΔspTaTLP1-VYNE-GW and ΔspTaTLP1-VYCE-GW was used as negative control. Positive control is MAPK9-AvrLm1.Yellow florescent protein (YFP) signal was visualized by confocal microscopy. Yellow fluorescence confirms protein-protein interaction due to complementation of the split YFP protein. **(B)** Co-expressed fluorescently tagged TaPR1 and TaTLP1 colocalized in the *N. benthamiana* leaf epidermal cells. The TaPR1-pEarlyGate103 protein fusion colocalized with TaTLP1-pGWB454 in *N. benthamiana* cells and Merge+ Bright field show yellow, prove TaPR1 and TaTLP1 are interaction. Scale bar, 50 μm. The red arrow showing TaTLP1 and TaPR1 were visualized in the apoplastic space (AP) as evidenced by the plasmolysis.

## TaPR1 stabilizes the accumulation of TaTLP1

ΔspTaTLP1-pGWB414 (HA-tagged TaTLP1), ΔspTaPR1-pEarlyGate103 (GFP-tagged TaPR1), and a Myc-tag negative control were co-expressed in *N. benthamiana* using agroinfiltration. Most protein degradation events are accomplished via the ubiquitin/26S proteasome pathway [32]. MG132, a proteasome inhibitor, was used to examine and compare the stability and accumulation of TaTLP1 to test whether it was regulated by this pathway. In the absence of MG132, ΔspTaTLP1-pGWB414 accumulated to a higher level in the presence of

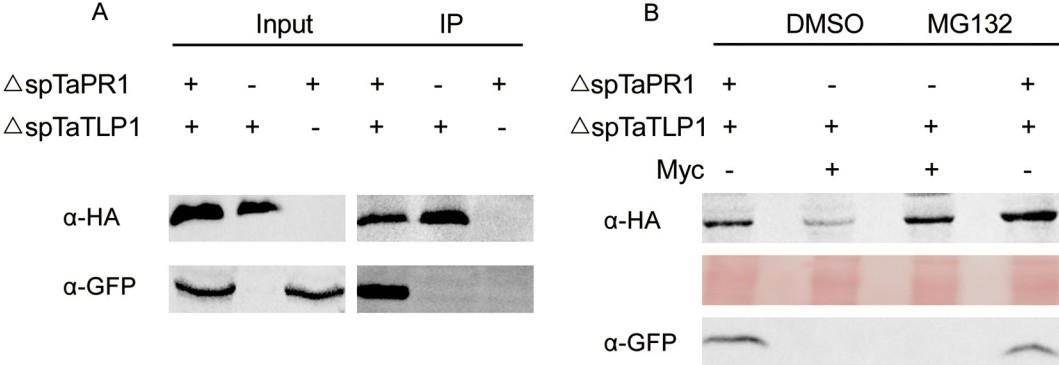

**Fig 5. TaPR1 stabilizes the accumulation of TaTLP1 and enhances resistance to wheat leaf rust. (A)** Co-immunoprecipitation of TaPR1 and TaTLP1 from the total plant protein extracts. ΔspTaPR1-pEarlyGate103 and ΔspTaTLP1-pGWB414 were co-expressed in *N. benthamiana*. Proteins were extracted after 48 h and subjected to immunoprecipitation by HA-magnetic beads. Immunoprecipitated proteins (IPs) were analyzed by immunoblotting and probing with either anti-HA (α-HA) or anti-GFP (α-GFP). **(B)** ΔspTaTLP1-pGWB414 and ΔspTaPR1-pEarlyGate103 were co-expressed in *N. benthamiana* using agroinfiltration, and Myc-tagged was used as negative control. Dimethyl sulfoxide (DMSO) or MG132 (100 mM) was infiltrated into the leaves at 24 hr post-infiltration. Proteins were extracted after 48 hr post-infiltration and subjected to western blotting.

ΔspTaPR1-pEarlyGate103 compared to the Myc control. However, *N. benthamiana* leaves co-expressing ΔspTaTLP1-pGWB414 andΔspTaPR1-pEarlyGate103 or Myc that were pretreated with MG132 resulted in increased accumulation of TaTLP1 in the leaves of control treatments (co-expressing ΔspTaTLP1-pGWB414 and Myc) to a similar level of accumulation of TaTLP1 that was observed in the leaves co-expressing ΔspTaTLP1-pGWB414 and ΔspTaPR1-pEarly-Gate103 (Fig 5B). Taken together, these results suggest that TaPR1 stabilizes TaTLP1 by preventing its degradation mediated by the 26S proteasome.

## A conserved 52-amino-acid region of TaPR1 is involved in its binding to TaTLP1

To define which region of TaPR1 is required for interaction with TaTLP1, a preliminary prediction of Protein-Protein Interaction strength was made using software SWISS-MODEL (S1 Fig), and two N-terminal and three C-terminal truncated TaPR1 protein variants were generated based on the TaPR1 protein secondary structure [15] (S2 Fig). A total of five variants were constructed including $\Delta N_{25-45}$-TaPR1-pGADT7, $\Delta N_{46-64}$-TaPR1-pGADT7, $\Delta C_{113-164}$-TaPR1-pGADT7, $\Delta C_{128-164}$-TaPR1-pGADT7, and $\Delta C_{143-164}$-TaPR1-pGADT7. Only $\Delta C_{113-164}$-TaPR1 failed to interact with TaTLP1, indicating that 52 amino acids located at C-terminal region of TaPR1 is required for the interaction, but the remaining region of TaPR1 is dispensable for interaction with TaTLP1 (Fig 6).

It has been reported that cysteines are important for TaTLP1 structure [33]. To further define the roles of disulfide bonds in the stabilization of the TaPR1-TaTLP1 interaction or in determining individual protein structures that confer interactions of TaTLP1, we generated two mutated TaTLP1 proteins with five cysteine to alanine mutations at the N-terminal region and another five cysteine to alanine mutations at the C-terminal region, and used these variants to perform Y2H experiments with TaPR1, respectively. Interestingly, one of the C-terminal cysteine-mutated regions weakened the growth of yeast on selective plates, indicating that mutation of C-terminal cysteines from TaTLP1 affects its interaction with TaPR1. Furthermore, we mutated the individual cysteines at the C-terminal, and we found that the effects are similar to the mutation of all cysteines in combination (S3 Fig), indicating that cysteines affect the interaction between TaTLP1 and TaPR1, but are not indispensable for the interaction of TaPR1 and TaTLP1.

## Antifungal activity of TaTLP1-TaPR1 is higher compared to either TaTLP1 or TaPR1 alone

A PR5-like protein previously has been reported to inhibit the growth of yeast and filamentous fungi, such as *Botrytis cinerea* [34]. In our previous study, we found that *TaTLP1* was required for *Lr35*-mediated resistance against leaf rust fungus via BSMV-induced gene silencing analysis [24]. However, it remains challenging to examine antifungal activity *in vitro* as leaf rust is an obligate biotroph, growing and reproducing exclusively on living host tissue. As an initial test to establish a possible role for TaTLP1-TaPR1 in protecting wheat against *Pt*, growth-inhibition assays were conducted. The pGEX-6P-3 vector system was used to express TaTLP1 in *Escherichia coli* with a protein tag (GST) for purification and subsequent antibody recognition (S4 Fig). We analyzed the germination of urediniospores, the length of hyphae, and the number of hyphal branches in the presence of pure TaTLP1-TaPR1, TaTLP1 alone, and TaPR1 alone, respectively, using sterile water, elution buffer and pGEX-6P-3 vector as a control. Microscopic observation showed that the germination of urediniospores and the growth of germtubes were significantly restricted with pure proteins (Fig 7, S2 Table). The antifungal activity of TaTLP1 or TaPR1 alone is significantly higher compared to controls, and the

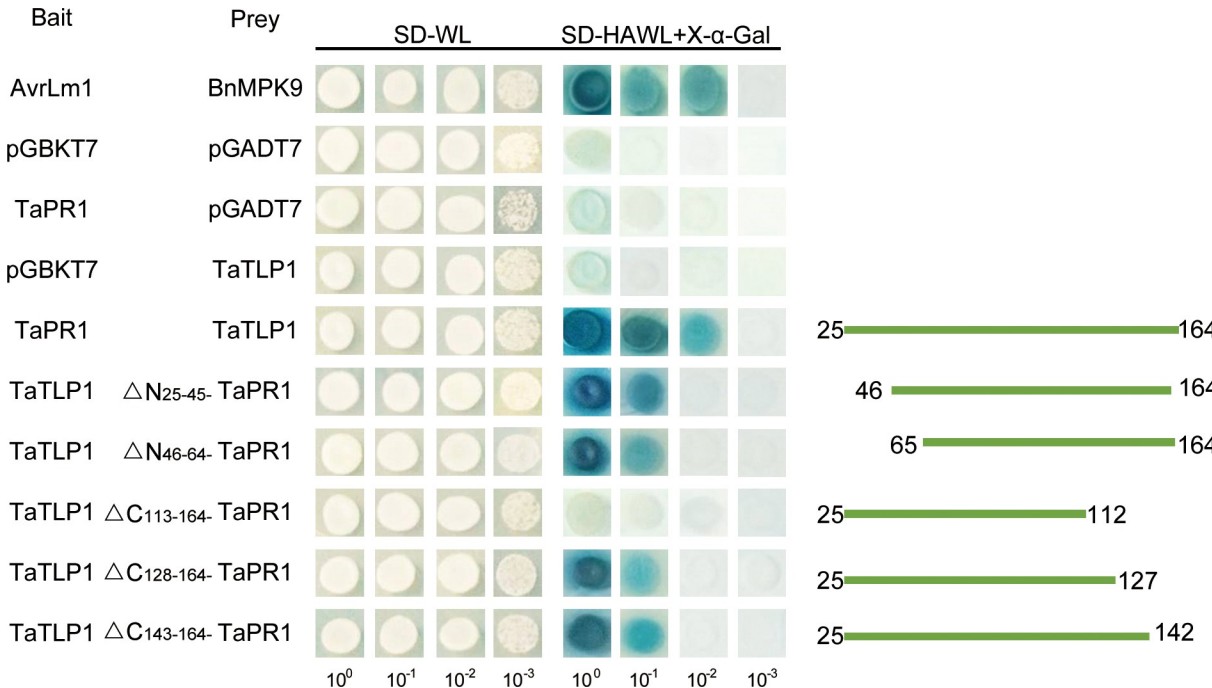

**Fig 6. A conserved 52-amino-acid region of TaPR1 is required for the interaction with TaTLP1.** Truncated TaPR1 constructs were generated including N-terminal deletion of residues 25 to 45 ($\Delta N_{25-45}$ -TaPR1), deletion of residues 46 to 64 ($\Delta N_{46-64}$ -TaPR1), and C-terminal deletion of residues 113 to 164 ($\Delta C_{113-164}$ -TaPR1), deletion of residues 128 to 164 ($\Delta C_{128-164}$ -TaPR1), deletion of residues 143 to 164 ($\Delta C_{143-164}$ -TaPR1). Each of the constructs was co-transformed with TaTLP1 into yeast. All transformants are able to grow on synthetic dropout medium without leucine and tryptophan (SD-WL) medium. Yeast colonies that were able to grow on selective medium (SD medium without leucine, tryptophan, histidine, and adenine supplemented with X-α-Gal and AureobasidinA [SD-HAWL]) and displayed blue coloration confirmed the protein-protein interaction. Positive control is MAPK9-AvrLm1. TaPR1 and pGADT7, TaTLP1 and pGBKT7, pGADT7 and pGBKT7 are used as the negative controls.

antifungal activity of TaTLP1-TaPR1 is higher compared to either TaTLP1 or TaPR1 alone, indicating that the proteins alone and especially in combination protect wheat by inhibiting the growth of leaf rust fungus and contribute to resistance against *Pt*.

## Co-silencing of *TaTLP1* and *TaPR1* significantly enhances wheat susceptibility

To further confirm the function of the interaction between TaTLP1 and TaPR1 in wheat resistance to *Pt*, the BSMV-virus-induced gene silencing (VIGS) system was employed to knock down the transcription of *TaTLP1* and *TaPR1*, respectively, and we analyzed the impact on wheat response to pathogen infection. Then, BSMV: TaTLP1-V2 and BSMV: TaPR1-V2 recombinant vectors were mixed together and rub-inoculated onto TcLr19 seedlings to detect the effect of *TaTLP1-TaPR1* co-silencing on wheat response to *Pt* infection. All BSMV-inoculated plants displayed mild chlorotic mosaic symptoms depicting infection by the virus (Fig 8A). Photo-bleached symptoms were observed on wheat leaves after inoculation with BSMV: TaPDS at 10 dpi when *TaPDS* was silenced (Fig 8A). After inoculation with fresh urediospores of the *Pt* race PHNT, a conspicuous hypersensitive response (HR) was elicited on leaves inoculated with BSMV: γ. Various numbers of *Pt* urediniospores were produced around the necrotic spots on the fourth and fifth leaves infected with BSMV: TaTLP1 and BSMV: TaPR1, as compared with BSMV: γ plants (Fig 8A). Silencing of both *TaTLP1* and *TaPR1* produced slightly larger pustules compared to the control, and more urediniospores were observed on the

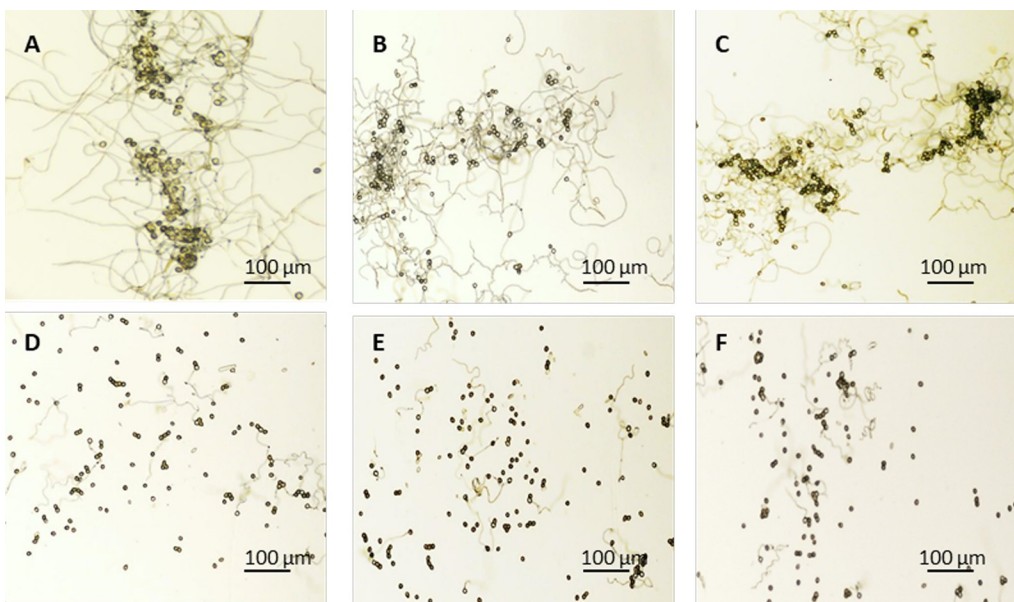

**Fig 7. The antifungal activity of TaTLP1-TaPR1 *in vitro*.** The sensitivity of urediospores to the presence of pure protein on the 2% agar medium under 22–25°C dark for 12 hours. **(A)** sterile water **(B)** elution buffer **(C)** pGEX-6P-3 vector **(D)** TaPR1 pure protein **(E)** TaTLP1 pure protein **(F)** TaTLP1-TaPR1 pure protein. Pure protein was used as the same concentration. Sterile water, elution buffer and pGEX-6P-3 vector was used as control, and photographs were taken at 10× magnification. Scale bar, 100 μm.

BSMV: TaTLP1/TaPR1 plant leaves than those with single gene silencing at 14 dpi with *Pt* race (Fig 8A).

To determine the silencing efficiency of *TaTLP1* and *TaPR1* in plants that had been infected with recombinant vectors, we detected the relative expression levels of *TaTLP1* and *TaPR1* in infected leaves using qPCR. Our results showed that the relative expression of *TaTLP1* was significantly reduced by 75%, 60.5%, and 69.3% after infection with avirulent *Pt* at 24, 48, and 120 hpi, respectively, whereas the corresponding *TaPR1* transcript expression in infected BSMV: TaPR1 leaves was reduced by 62.4%, 61.7%, and 57.58%, respectively (Fig 8B). Approximately 77.4%, 63.96%, and 72.8% of the *TaTLP1* transcripts and 72.5%, 67.6%, and 61.7% of the *TaPR1* transcripts were suppressed for each respective time point in the *TaTLP1/TaPR1* co-silenced plants compared with that in BSMV: γ inoculated plants, suggesting that both of *TaTLP1* and *TaPR1* were silenced successfully (Fig 8B). The silencing efficiency further confirmed that the enhanced susceptibility phenotypes observed on the leaves inoculated with the avirulent *Pt* race were due to the silencing of *TaTLP1* and *TaPR1*.

The histological changes in both the susceptible control (Thatcher) and silenced plants infected with *Pt* were observed. The number of hyphal branches in Thatcher was higher than in all BSMV silenced plants at 24 hpi (Fig 9A and 9B), and the hyphal length was longer in Thatcher than all knock-down plants (Fig 9C). At 120 hpi, the hyphal branches, the hyphal length, and the infection areas increased in different plants, and *TaTLP1/TaPR1*-knockdown plants displayed more hyphal branching, larger infection areas, and longer hyphal length compared to the control group (BSMV: γ) at 120 hpi (Fig 9A–9D). These results indicate that co-silencing of *TaTLP1/TaPR1* compromises the resistance response to *Pt* infection.

High concentration of ROS induces cell death in plants [35]. $H_2O_2$ is accumulated in different defense responses during wounding, insect attacks, and pathogen infections [36]. $H_2O_2$ accumulation was evaluated to further understand how *TaTLP1/TaPR1* participates in wheat

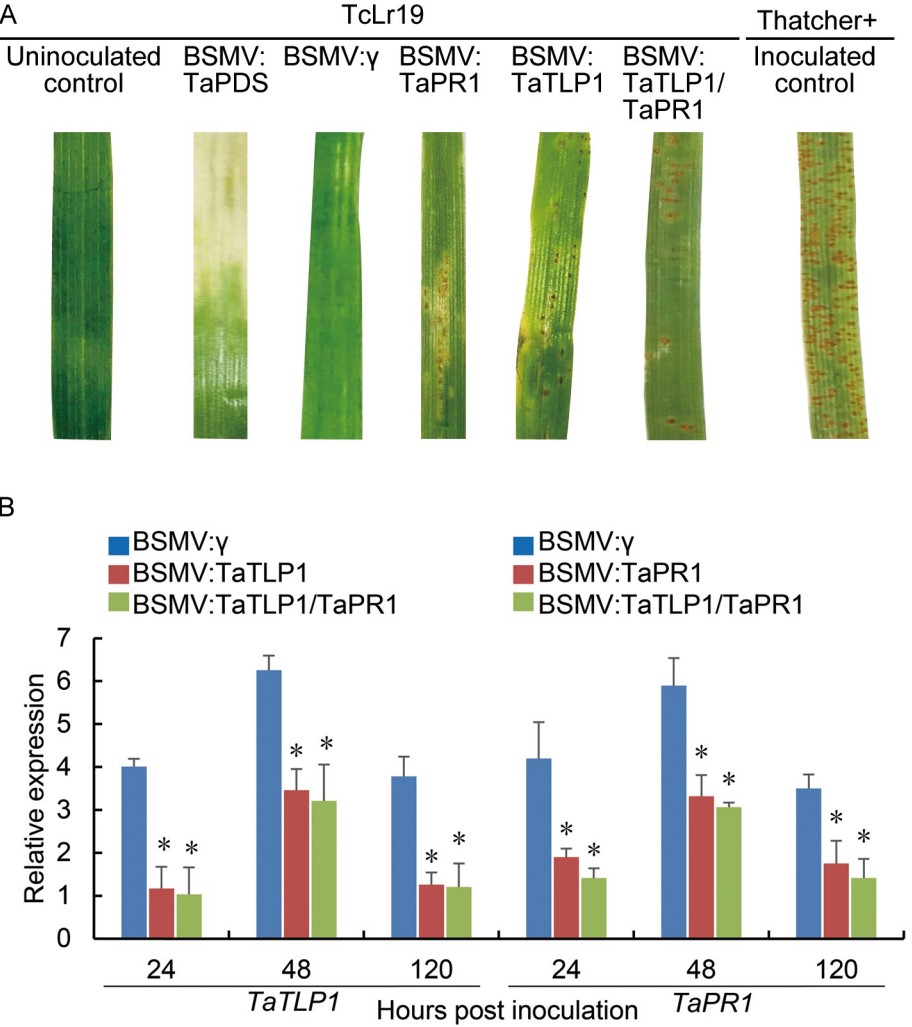

**Fig 8. Co-silencing of *TaTLP1* and *TaPR1* significantly enhances wheat susceptibility. (A)** Photos of the fourth leaves of TcLr19 wheat inoculated with urediospores after infection with BSMV: γ, BSMV: TaPDS, BSMV: TaTLP1, BSMV: TaPR1, and BSMV: TaTLP1/TaPR1 at 14 dpi. BSMV: γ, BSMV: TaPDS, inoculated Thatcher and uninoculated leaves were used as positive and negative control, respectively. **(B)** Relative transcript levels of *TaTLP1* and *TaPR1* in control and knockdown plants individually in *TaTLP1*-silenced plants or *TaPR1*-silenced plants or simultaneously in *TaTLP1/TaPR1*-silenced plants at 24, 48 and 120 hpi. The y-axis indicates the amounts of *TaPR1* or *TaTLP1* transcript normalized to the *GAPDH* gene. The x-axis indicates sampling times. The transcript levels of *TaPR1* and *TaTLP1* in BSMV: γ were standardized as 1. Data are means ± standard errors (SE) of three independent experiments. Differences between that in knockdown plants and control plants were assessed using Student's *t*-tests (* $p < 0.05$).

resistance. The areas of cell death and $H_2O_2$ accumulation were significantly lower at 120 hpi in *TaTLP1/TaPR1*-knockdown plants than that observed in BSMV: γ, *TaTLP1* alone, or *TaPR1* alone inoculated leaves (Fig 10). To further confirm the decrease in ROS accumulation in silenced plants and to understand the mechanism behind the decreased oxidative stress tolerance, the expression of genes involved in ROS signaling, catalase (*TaCAT*) and superoxide dismutase (*TaSOD*), were assayed. As shown in S5 Fig we observed an increase in the accumulation of *TaCAT* and *TaSOD* transcripts in *TaTLP1*, *TaPR1*, and especially *TaTLP1/TaPR1*-knockdown plants compared to BSMV: γ control plants. These observations suggest that *TaTLP1* and *TaPR1* may have synergistic effects on their respective functions (S5 Fig).

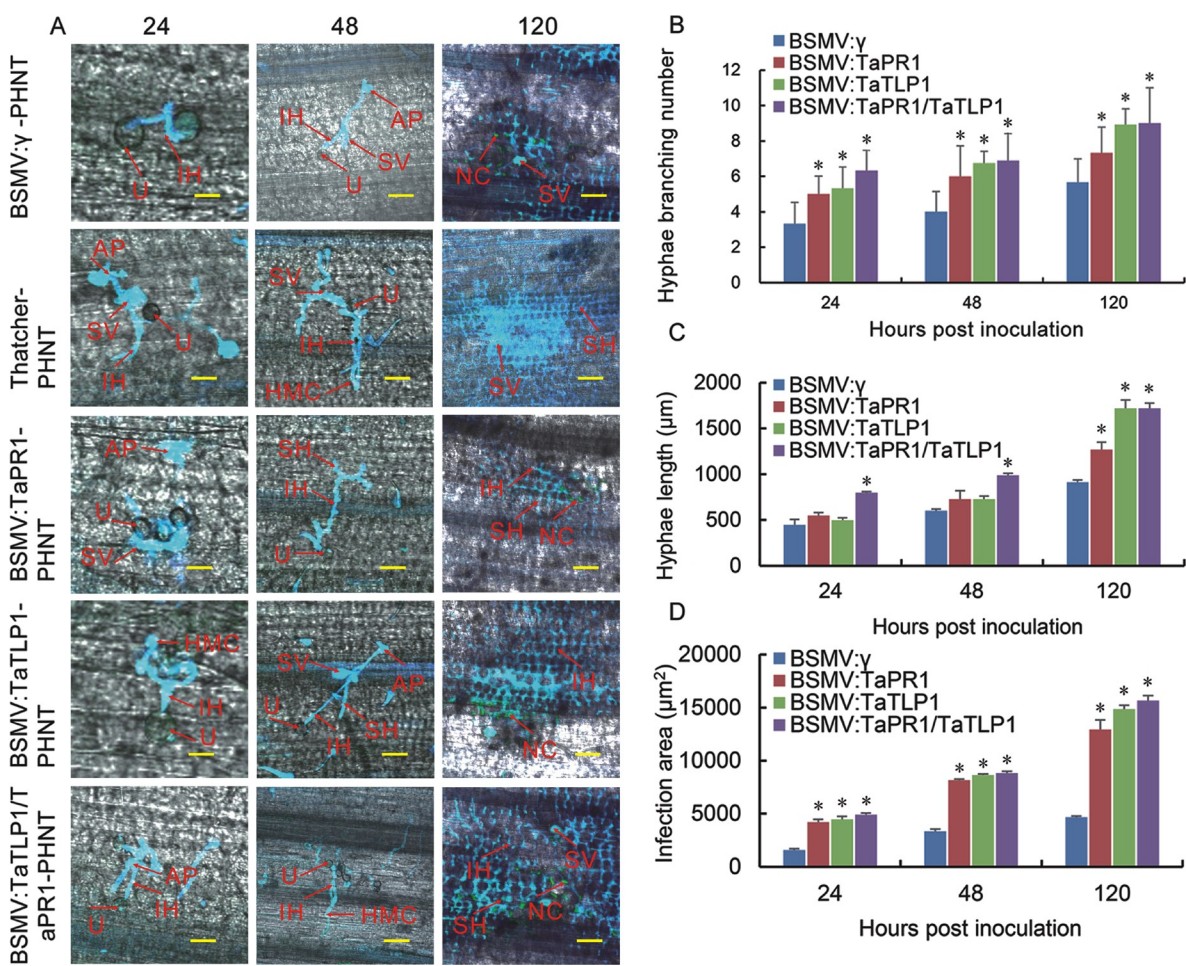

**Fig 9. Histological changes in both susceptible control (Thatcher) and silenced plants infected with *Pt* were observed. (A)** Histological observation of hypha development and host cell death in wheat leaves BSMV-infected wheat leaves after inoculation with *Pt* at 24, 48 and 120 hpi. AP: appresorium; SV: substomatal vesicle; IH: infection hypha; HMC: haustorial mother cell; SH: second hypha; NC: necrotic cell; U: urediospore. Scale bar, 50 μm. **(B, C, D)** The fungal growth including the the number of mycelial branches, length of the hyphae, the area of infection, and of *Pt* in wheat leaves inoculated with BSMV: γ, BSMV: TaPR1, BSMV: TaTLP1 and BSMV: TaPR1-TaTLP1 at 24, 48, and 120 hpi was observed. The fungal growth level of BSMV: γ was standardized as 1. Data are means ± standard errors (SE) of three independent experiments. Differences were assessed using Student's *t*-tests (* $p < 0.05$).

## Discussion

PR proteins are encoded pathogenesis-induced genes and play a supporting role in plant defense and the accumulation of hormones associated with plant defense. However, the function of the PR family members and the coordination between them remains unclear. *TLP* is involved in plant defense against various biotic and abiotic stresses [5]. The evolution of the *TLP* gene family was investigated by performing a genome-wide identification and comparison in *Arabidopsis*, *Oryza*, *Populus*, *Zea*, *Physcomitrella* and *Chlamydomonas*, and functional network analysis demonstrated that some resistance genes work together with *TLPs* [7]. As we all know, *TLPs* and *PR1* are often considered to be common marker genes for antipathogen responses and were also found to interact with pathogen receptors in recent years [8–10]. In our previous study, we reported that *TaPR1* and *TaTLP1* play important roles in wheat development and resistance to leaf rust pathogen attack [24, 29]. Interestingly, now we identified TaPR1 as an interacting partner of TaTLP1. Although there are many reports of *PR* genes

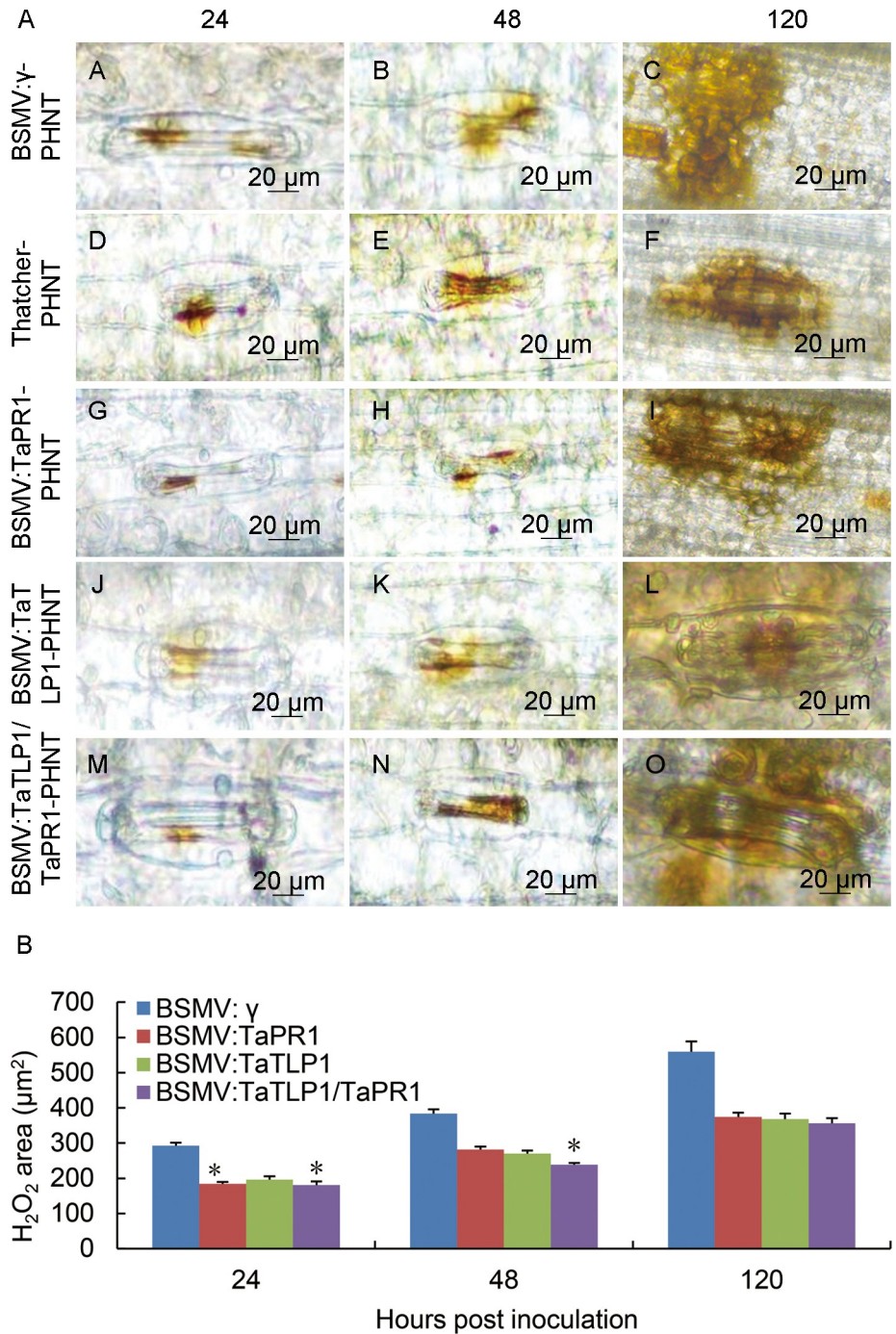

**Fig 10. $H_2O_2$ production and necrosis were observed in these leaves at 24, 48 and 120 hpi. (A)** $H_2O_2$ accumulation at infection sites was detected by staining with 3, 3-diaminobenzidine (DAB) and viewed under differential interference contrast optics. Scale bar, 20 μm. **(B)** The accumulation of $H_2O_2$ was measured by calculating the DAB-stained area at each infection site at 24, 48, 120 hpi. Image J was employed to quantify $H_2O_2$ area. A minimum of 50 infection sites were examined. The $H_2O_2$ area of BSMV: γ was standardized as 1. Data are means ± standard errors (SE) of three independent experiments. Differences were assessed using Student's *t*-tests (* $p < 0.05$).

involved in plant pathogen defense, as far as we know, this is the first report of the interaction of two PR molecules in wheat, which sheds some light on the mechanism of *PR* genes in plants and might provide a basis for further functional investigation of these gene families.

A PR1-derived defence signalling peptide from the C-terminus of TaPR1-1, known as CAPE1, enhanced the infection of wheat by *Parastagonospora nodorum* in an SnTox3-dependent manner, but played no role in ToxA-mediated disease [15]. The CAPE1 peptide with a PxGNxxxxxPY- motif was derived from the C-terminal end of tomato PR-1b by a novel peptidomics approach and shown to increase immunity against pathogen infection in tomato and *Arabidopsis*. It was speculated that the CNYx motif immediately upstream of CAPE1 is required for cleavage from PR-1b [37]. In this study, we tested to see if the interaction of TaPR1-TaTLP1 required CAPE1. The CNYx motif and CAPE1 peptide consists of 11 amino acids (PxGNxxxxxPY-) encoded within the CTR (S2 Fig) of TaPR1 (amino acids 143–164) were mutated and showed that CAPE1 appears to not be required for interaction of TaPR1-TaTLP1 (Fig 6). These results are different in that a mutation of CNY sequence can abolish the cleavage of the AtCAPE1 peptide [38], and the peptide sequence shows that it appears conserved in most of the wheat PR1 proteins (S2 Fig). However, 52 amino acids located at C-terminal region of TaPR1 are required for its interaction with TaTLP1 (Fig 6). Precipitating from this work, we identified that a total of 15 amino acids from 113 to 127 located at C-terminal region of TaPR1 is required for the interaction. Most TLPs have 10 or 16 conserved cysteine residues, which might form eight disulfide linkages to stabilize proteins and resist unfavorable pH, proteases, and heat-induced denaturation [33, 39]. Thus, we investigated the function of cysteine residues in the interaction between TaTLP1 and TaPR1. We found that cysteine is not completely indispensable for the interaction of TaPR1 and TaTLP1, though it can affect the interaction between TaTLP1 and TaPR1. We will find the particular region of TaTLP1 which might be required for the interaction with TaPR1 in the future.

Leaf rust forms an intimate connection with the host through the formation of haustoria, which emerge from extracellular pathogen infection structures and enter plant cells through the infection peg by invaginating the host plasma membrane [40], so the interaction between host and fungus happens in the space between the cell membrane and cell wall. Consistent with this scheme for pathogenesis, our co-localization results showed that TaTLP1 interacts with TaPR1 in the plant cell apoplast (Fig 4B). Based on the presented data, we investigated the stability and level of accumulation of TaTLP1 in the presence of TaPR1. The results suggest that TaPR1 potentially increases the stability of TaTLP1. Moreover, to our surprise, it showed that pure TaTLP1-TaPR1 complexes play a crucial role in suppressing the germination of urediniospores and hyphal growth which indicated that the interaction of TaTLP1 and TaPR1 contribute to wheat defense responses to *Pt*.

*In vivo*, BSMV-VIGS was conducted to co-silence *TaTLP1* and *TaPR1* individually or simultaneously. We have reported that BSMV-induced *TaTLP1* gene silenced wheat plants exhibited obviously compromised resistance, which suggests that *TaTLP1* is involved in *Lr35*-mediated wheat defense in response to leaf rust attack [24]. However, characterization of PR-1 function is a challenging task because of the genomic redundancy of the PR-1 family. Lu reported that attempts to specifically knock down the *PR-1-5* gene by VIGS were unsuccessful [14]. In this study, based on the identification of the functional regions and off-target prediction of *TaPR1*, two fragments of *TaPR1* were knocked down successfully and showed be involved in wheat defense in response. In addition, we found that the silencing efficiency of TaPR1-V2 is higher than TaPR1-V1 based on phenotype, qPCR analysis, histological observations of fungal growth and $H_2O_2$ accumulation (S6 Fig, S7 Fig). Thus, TaPR1-V2 was selected to detect the co-silencing of *TaTLP1* and *TaPR1* genes by the VIGS system and verified the function of their interaction. TaPR1 belongs to PR1-4 and shares 94.51% homology with PR1-

5 (S2 Fig), we detected the copy number of *TaPR1* by Southern blotting and found that there are 4 copies in the wheat genome which is consistent with BLAST analyses (S8 Fig, S9 Fig). Meanwhile, *TaPR1* was located on 7D by chromosome localization using Chinese Spring and its null-tetrasomic lines (S10 Fig). A VIGS fragment was designed according to the specific region of *TaPR1* so that it can represent the function of the gene (S6 Fig). Similar to *Lr21* [41], in the case of *TaTLP1*, we have reason to believe that the level of suppression may actually be significantly greater than what was measured in this analysis. As for *TaTLP1*, it also has 3 copies located at 7A, 7B and 7D respectively, and their sequences are almost identical, so it is difficult to distinguish them and to identify a single polymorphic nucleotide that could be used to specifically measure independent gene expression (S11 Fig). Therefore, qPCR experiments likely measure the combined expression of *TaTLP1* and its homologs.

Taken together, our findings indicate that the TaTLP1-TaPR1 complex positively contributes to wheat resistance to leaf rust and regulates ROS generation in the interaction between wheat and *Pt* (S12 Fig), which will broaden our understanding of the potentially versatile functions of PR proteins and provide valuable insight into the molecular mechanism of plant immunity. Because VIGS is a transient expression system, we found that the levels of expression for *TaPR1* and *TaTLP1* are higher at 48 hpi compare to 24 hpi and then decrease again at later time point after inoculation with *Pt*, which is similar to what was found during pathogenesis by stripe rust [42]. In addition, under diseased conditions where there are many proteins which are differentially expressed, the complex responses to pathogenesis can alter binding affinity between these two proteins. In the future, we plan to evaluate the resistance ability of transgenic plants combining both two PR proteins because it is likely that PR proteins may be effective only in combination with select other PR proteins [43].

## Materials and methods

### Plant materials and leaf rust isolate

Wheat seedlings of cultivar TcLr19 (Tc*6/RL6040), which is a near-isogenic line containing the *Lr19* gene, was requested from the Cereal Disease Lab of USDA located at University of Minnesota were preserved in the Laboratory of Leaf rust, Hebei Agricultural University. *Pt* pathotype 07-10-426-1 (PHNT), collected from China and preserved in our laboratory was used to inoculate wheat according to Roelfs' standard [44]. For plant extraction, second leaves inoculated with PHNT or sterile distilled water (control) were harvested at 0, 24, 48, 72, 96, 120, 144, 168 hours post inoculation (hpi). All samples were immediately frozen in liquid nitrogen after harvesting and stored at -80˚C. Each treatment included three independent biological replicates.

### Binary vector construction, *Agrobacterium*-mediated transformation, and transient transformation assays

For transient expression, full length *TaTLP1* was PCR-amplified from wheat infected with *Pt* using primers TaTLP1-pENTR-F and TaTLP1-pENTR-R. The truncated *TaTLP1* lacking the signal peptide sequence was amplified using primers ΔspTaTLP1-pENTR-F/ΔspTaTLP1-pENTR-R. The PCR products were introduced into entry vector pENTR-D-TOPO (Invitrogen) to obtain TaTLP1-pENTR and ΔspTaTLP1-pENTR which were recombined into the binary vector pEarlyGate103 (with green fluorescence protein (GFP) tag) and resulted in constructs TaTLP1-pEarlyGate103 and ΔspTaTLP1-pEarlyGate103 respectively. *Agrobacterium*-mediated transient expression in *N. benthamiana* was performed according to methods described previously [45]. The truncated *TaPR1* lacking the signal peptide sequence was

amplified using primers ΔspTaPR1-pENTR-F/ΔspTaPR1-pENTR-R and the PCR products were introduced into entry clone pENTR-D-TOPO (Invitrogen) to obtain ΔspTaPR1-pENTR. All pENTR clones were confirmed by sequencing.

To generate the BiFC constructs, ΔspTaTLP1-pENTR and ΔspTaPR1-pENTR were recombined into the binary vectors pDEST-VYNE-GW and pDEST-VYCE-GW, respectively [31]. The binary vector TaTLP1-pGWB454 (with RFP tag) was constructed for co-localization assays. All of these resulting plasmids were used for *Agrobacterium*-mediated transformation.

For CoIP, ΔspTaTLP1-pENTR was recombined into the binary vector pGWB414 containing the C-terminal Hemagglutinin (HA) tag fused to *TaTLP1*. ΔspTaPR1-pENTR was recombined into the binary vector pEarlyGate103 containing the C-terminal GFP tag fused to TaPR1. The resulting plasmids ΔspTaPR1-pEarlyGate103 together with construct ΔspTaTLP1-pGWB414 were transformed into *A. tumefaciens* strain GV3101.

## Protein extraction, pull-down experiments and protein identification

Proteins were extracted from *N. benthamiana* leaves 48 h after infiltration with a mixture of *A. tumefaciens* GV3101 containing corresponding constructs as previously described [46]. For pull-down assays, Chromotek GFP-Trap was used to bind GFP-tagged protein, Anti-GFP Affinity Gel beads were added into the protein mixture and the manufacturer's instructions were followed. The GFP-Trap products were separated by 15% SDS-PAGE and visualized by coomassie blue staining, then transferred onto membranes (NC) using wet blotting overnight. BSA (3%) was used as a blocking agent. A 1:1000 dilution of anti-GFP (Pierce) was used. The secondary antibody goat-anti-mouse (Pierce) was used at a 1:1000 dilution. The luminescent signal was visualized by Immobilon Western Chemiluminescent HRP Substrate using the ChemiDoc Touch Imaging System (Bio-Rad). The pull-down and western blot assays were repeated at least twice. Different bands in TaTLP1-pEarlyGate103 extraction were compared to control bands from pEarlyGate103 empty vector extraction, cut off from the gel, and then subjected to mass spectrometry entrusted to the Beijing Genomics Institute (BGI).

## RNA isolation and quantitative PCR

Total RNA was extracted using the TaKaRa MiniBEST Universal RNA Extraction Kit (TaKaRa) according to the manufacturer's instructions. Contaminating DNA was digested with amplification-grade DNase I (TaKaRa) and cDNA was synthesized using the SuperScript II reverse transcriptase, oligo (dT)-18T, and RNase Inhibitor (TaKaRa). qPCR was performed with the LightCycler 96 (Roche). Expression of *TaTLP1* and *TaPR1* were investigated, and the wheat glyceraldehyde-3-phosphate dehydrogenase (*GAPDH*, GenBank accession No. AF251217) gene was used to calibrate the expression level of the query genes, as previously described [29] (S3 Table). In addition, the expression levels of superoxide dismutase (*TaSOD*, GenBank accession number CB307850) and catalase (*TaCAT*, GenBank accession number X94352) genes were analyzed by qPCR in order to confirm reduction of ROS accumulation in silenced plants. Quantification of the target gene was assessed by relative standard curves. The $2^{-\Delta\Delta CT}$ method [47] was employed to quantify the relative gene expression levels. Statistical analysis Microsoft Excel software was used to calculate mean values and standard errors (SE). The statistical significance of differences was calculated using Student's *t*-test by the SPSS 21.0 (IBM SPSS Statistic) to obtain the P-value. Data were shown as mean ± SE of three biological replicates from one representative experiment. Significant differences between treatments and controls or between time-course points were represented by asterisk ($^*p < 0.05$).

## Yeast two-hybrid assays

The Matchmaker Gold Yeast Two-Hybrid System (Clontech) was used to verify the interaction between TaTLP1 and TaPR1. *TaPR1* (without signal peptide) was introduced into pGBKT7 as bait, while *TaTLP1* (without signal peptide) was introduced into pGADT7 as prey. The bait and prey plasmids were co-transformed into yeast strain Y187 according to the manufacturer's instructions. Truncated *TaPR1* mutants with different deletions in the N- or C- terminal region were generated by PCR amplification and were subcloned into pGADT7 (S3 Table). Cysteines were mutated to alanines in TaTLP1 by polypeptide synthesis. Transformed yeast cells were assayed for growth on synthetic dropout SD/-Trp-Leu plates and SD/-Trp-Leu-His-Ade plates containing X-α-Galactosidase (X-α-Gal) and Aureobasidin A (AbA).

## BiFC and co-localization assays

A gateway vector system enabling large-scale combination and investigation of candidate proteins was used for BiFC analysis as described previously [31]. Protein expression was confirmed by western blotting using the appropriate antibody. Then, plasmids of ΔspTaTLP1-VYNE-GW and ΔspTaPR1-VYCE-GW were co-transformed into *N. benthamiana*. The interaction of AvrLm1 with BnMPK9 was used as the positive control [48]. Fluorescence in leaves of *N. benthamiana* was monitored at 2 d after agroinfiltration, and then imaged directly using a confocal laser scanning microscope (CLSM) (Olympus FluoView FV1000).

## Co-IP assay

To validate the protein interactions of TaTLP1 and TaPR1, Co-IP assays were performed as follows. Total protein from *N. benthamiana* leaves 48 h after infiltrating with mixture of *A. tumefaciens* GV3101 containing either ΔspTaPR1-pEarlyGate103 or ΔspTaTLP1-pGWB414 were isolated by homogenizing tissues with RIPA buffer 3 d after agroinfiltration, and were subjected to α-HA IP. Eluted proteins were subjected to immunoblot analysis with anti-GFP antibody. Approximately 25 μL of RIPA buffer containing the total proteins was loaded as an input control.

## Antifungal activity assays

*TaTLP1* and *TaPR1* were cloned into the *BamH* I and *EcoR* I restriction sites of plasmid pGEX-6P-3 to generate the plasmid TaTLP1-GST, TaPR1-GST, respectively. The recombinant plasmids were transformed into *E. coli* Top10 for expression. Proteins were detected by SDS-PAGE, and were frozen at −80˚C in small aliquots until use. The *Pt* strains PHNT were tested for their response in the presence of pure TaTLP1-TaPR1 complexes, TaTLP1 alone and TaPR1 alone respectively. According to the experiment used to detect germination of urediniospores, petri dishes were filled with 15ml agar medium containing purified TaTLP1 or TaPR1 (1mg/ml), or TaTLP1 plus TaPR1 (1mg/ml). A final concentration of 116.67 ml$^{-1}$ spore of PHNT was added into the agar medium. The plates were incubated at 25℃ in the dark and the germination of spores, hyphal length, and branches were observed with the Nikon Ti2-LAPP Ti2 Laser Application System (Nikon Corporation). The same experiment was also carried out to evaluate growth inhibition with sterile water, buffer instead of the protein solution used as negative controls.

## Virus-induced gene silencing (VIGS)

In order to confirm the specificity of the silenced fragments, fragments were selected that had no or lower similarity with other genes to prevent off-target, then one cDNA fragments of

*TaTLP1* and two cDNA fragment of *TaPR1* for gene silencing with *Not* I and *Pac* I restriction sites (S3 Table) were obtained by reverse transcription PCR (RT-PCR) and inserted into the original barley stripe mosaic virus (BSMV) vector [49]. *In vitro* transcription was performed using the mMESSAGE mMACHINE Kit High Yield Capped RNA Transcription Kit (Ambion) with linearized plasmid as template according to the manufacturers protocol. Then 240 μl of the BSMV mixture was applied to fully expanded leaves of 2-leave stage TcLr19 wheat plants by rubbing according to the method described previously [50]. Leaf rust uredin-iospores were inoculated on the fourth and fifth leaves of TcLr19 and the susceptible wheat cultivar Thatcher. Sterile water was inoculated to wheat leaves as control. The more details of VIGS assay were performed according to a previous study [24]. The feasibility and silencing efficiency were tested using the wheat phytoene desaturase (*TaPDS*) as a positive control.

### Histological observations of fungal growth and host responses

Auto-fluorescence of attacked mesophyll cells was observed in necrotic areas by epifluorescence microscopy (excitation filter, 488 nm; dichromic mirror, 510 nm; and barrier filter, 520 nm). 3, 3-Diaminobenzidine (DAB; Coolaber) staining was conducted to detect $H_2O_2$ accumulation following the protocols described previously [51] and was then viewed by differential interference contrast optics. A minimum of 50 infection sites were examined on each of five randomly selected leaf segments for every treatment. The hyphal length, the branching of hyphae and $H_2O_2$ accumulation were observed with the Nikon Ti2-LAPP Ti2 Laser Application System (Nikon Corporation).

### Supporting information

**S1 Fig. Prediction of protein secondary structure.** The structure of TaTLP1 **(A)** and TaPR1 **(B)** were shown as diagram and surface views. The N-terminal region, C-terminal region and $\Delta C_{113-164}$ are shown in these proteins structure of TaTLP1 and TaPR1. Online software SWISS-MODEL was used to predict these proteins the second structures.
(TIF)

**S2 Fig. Related to Fig 5 protein sequences alignment about TaPR1 and the other homologous proteins.** Sequence from NCBI (https://www.ncbi.nlm.nih.gov/). Gene ID: *TaPR1* from our previous study, ID: HQ848391, *PR1-1*'s gene ID: AJ007348, *PR1-2*'s gene ID: HQ541962, *PR1-3*'s gene ID: HQ541963, *PR1-4*'s gene ID: HQ541964, *PR1-5*'s gene ID: HQ541965. TaPR1 is consistent with PR1-4 sequence. Red box indicated signal peptide and CAPE1.
(TIF)

**S3 Fig. C-terminal cysteines from TaTLP1 affects its interaction with TaPR1.** Two mutated TaTLP1 proteins with five cysteines to alanines at N-terminal region (ΔNcys-TaTLP1) and another five cysteines to alanines at C-terminal region (ΔCcys-TaTLP1), and four individual mutated cysteine at C-terminal (ΔCcys1-TaTLP1, ΔCcys2-TaTLP1, ΔCcys4-TaTLP1, ΔCcys5-TaTLP1) that were used for Y2H with TaPR1 respectively. All transformants are able to grow on synthetic dropout medium without leucine and tryptophan (SD-WL) medium. Yeast colonies that were able to grow on selective medium (SD medium without leucine, tryptophan, histidine, and adenine supplemented with X-α-Gal and AureobasidinA [SD-HAWL]) and displayed blue coloration confirmed the protein-protein interaction. Positive control is MAP-K9-AvrLm1. TaPR1 and pGADT7, TaTLP1 and pGBKT7, pGADT7 and pGBKT7 are used as the negative controls.
(TIF)

**S4 Fig. The expression and purification of targeted protein. (A)** Map of plasmid pGEX-6P-3. *TaTLP1* and *TaPR1* were cloned into the *BamH* I and *EcoR* I restriction sites of plasmid pGEX-6P-3 to generate the plasmid TaTLP1-GST, TaPR1-GST. The recombinant plasmid contains a GST tag. **(B)** Detection of purified proteins by SDS-PAGE. Lanes 1, 2 and 3 show pGEX-6P-3, TaTLP1-GST, TaPR1-GST expressed by *E. coli* Top10, respectively. The proteins were stained with Coomassie Brilliant Blue. M: Protein marker. The sites of restriction endonucleases are *BamH* I and *EcoR* I. The black arrow showing 18 kDa indicated the TaTLP1-GST and TaPR1-GST respectively; The white arrow showing 25 kDa indicated GST.
(TIF)

**S5 Fig. Detection of *TaSOD* and *TaCAT* expression in gene silencing plants.** The y-axis indicates the amounts of *TaSOD* or *TaCAT* transcript normalized to the *GAPDH* gene. The x-axis indicates sampling times. The transcript levels of *TaCAT* and *TaSOD* in BSMV: γ were standardized as 1. Data are means ± standard errors (SE) of three independent experiments. Differences between that in knockdown plants and control plants were assessed using Student's *t*-tests (* $p < 0.05$).
(TIF)

**S6 Fig. Host response in *TaPR1*-knockdown plants challenged by *Pt* PHNT. (A)** Disease development in different silenced leaves after inoculation with leaf rust. Fourth leaves after infection with BSMV: TaPR1-V1, and BSMV: TaPR1-V2 at 14 dpi. BSMV: γ, BSMV: TaPDS, inoculated Thatcher and uninoculated leaves were used as negative and positive control, respectively. **(B)** Relative transcript levels of *TaPR1* in knockdown plants at 24, 48 and 120 hpi. The y-axis indicates the amounts of *TaPR1* transcript normalized to the *GAPDH* gene. The x-axis indicates sampling times. The transcript levels of *TaPR1* in BSMV: γ was standardized as 1. Data are means ± standard errors (SE) of three independent experiments. Differences between that in knockdown plants and control plants were assessed using Student's *t*-tests (* $p < 0.05$).
(TIF)

**S7 Fig. Histological observations of fungal growth in *TaPR1*-knockdown plants challenged by *Pt* PHNT. (A)** Histological observation of hypha development and host cell death in wheat leaves BSMV-infected wheat leaves after inoculation with *Pt* at 24, 48 and 120 hpi. Scale bar, 50μm. $H_2O_2$ accumulation at infection sites was detected. Scale bar, 20 μm. **(B)** Detection of *TaSOD* and *TaCAT* expression in gene silencing plants. The y-axis indicates the amounts of *TaSOD* or *TaCAT* transcript normalized to the *GAPDH* gene. The x-axis indicates sampling times. The transcript levels of *TaCAT* and *TaSOD* in BSMV: γ were standardized as 1. Data are means ± standard errors (SE) of three independent experiments. Differences between that in knockdown plants and control plants were assessed using Student's *t*-tests (* $p < 0.05$). **(C, D, E)** The fungal growth including the length of the hyphae, the area of infection, and the number of mycelial branches of *Pt* in wheat leaves inoculated with BSMV: γ, BSMV: TaPR1-V1 and BSMV: TaPR1-V2 at 24, 48, and 120 hpi was observed. The fungal growth level of BSMV: γ was standardized as 1. Data are means ± standard errors (SE) of three independent experiments. Differences were assessed using Student's *t*-tests (* $p < 0.05$).
(TIF)

**S8 Fig. The specificity analysis of VIGS fragments among all *TaPR1* members.** Multiple sequence alignment of the coding sequences for the four *TaPR1* copies was performed by BioEdit software. The fragments for VIGS are indicated by red box (TaPR1-V1) and black box (TaPR1-V2), respectively.
(TIF)

**S9 Fig. The result of *TaPR1* Southern blot.** 1–2: TcLr19 DNA was digested with *Hind* III, *Sac* I respectively.
(TIF)

**S10 Fig. Amplification results of *TaPR1* in Chinese spring Nulli-tetrasomic lines.** M: DL2000 Marker; 1–21: Nulli-tetrasomic lines respectively lacking 1A, 1B, 1D, 2D, 2A, 2B, 3D, 3B, 3A, 4B, 4D, 4A, 5D, 5B, 5A, 6D, 6B, 6A, 7A, 7D, 7B chromosome. *TaPR1* is located on 7D.
(TIF)

**S11 Fig. Multiple sequence alignment of the coding sequences for the three *TaTLP1* copies.**
(TIF)

**S12 Fig. TaTLP1-TaPR1 complex positively contributes to wheat resistance to leaf rust pathogen.** Upon *Pt* infection wheat, a receptor localized in plant plasma membrane recognizes an unknown molecule of *Pt*, activating cellular signaling PTI and producing TaTLP1 and TaPR1. TaTLP1 could interact with TaPR1 and protect the wheat against fungal action, and regulated ROS generation in the interaction between wheat and *Pt*.
(TIF)

**S1 Table. Proteins identified in MS analyses after GFP-Trap.**
(DOCX)

**S2 Table. Antifungal activity of TaTLP1-TaPR1.**
(DOCX)

**S3 Table. List of primers used in this study.**
(DOCX)

## Acknowledgments

We would like to thank Professor M. Hossein Borhan from Agriculture and Agri-Food Canada, for providing the gateway vectors.

## Author Contributions

**Conceptualization:** Haiyan Wang.

**Data curation:** Fei Wang, Shitao Yuan.

**Formal analysis:** Fei Wang, Shitao Yuan, Wenyue Wu.

**Funding acquisition:** Haiyan Wang.

**Investigation:** Shitao Yuan, Yiqing Yang.

**Methodology:** Fei Wang, Wenyue Wu, Zhongchi Cui.

**Project administration:** Fei Wang, Haiyan Wang.

**Resources:** Haiyan Wang, Daqun Liu.

**Software:** Fei Wang, Shitao Yuan.

**Supervision:** Haiyan Wang, Daqun Liu.

**Validation:** Haiyan Wang.

**Visualization:** Fei Wang, Zhongchi Cui.

**Writing – original draft:** Fei Wang.

**Writing – review & editing:** Haiyan Wang.

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
