## [Decision Letter · Decision Letter 0]

29 Oct 2019

Dear Dr Wang,

Thank you very much for submitting your Research Article entitled 'TaTLP1 interacts with TaPR1 to contribute to wheat defense responses to leaf rust fungus' to PLOS Genetics. Your manuscript was fully evaluated at the editorial level and by independent peer reviewers. The reviewers appreciated the attention to an important problem, but raised some substantial concerns about the current manuscript. Based on the reviews, we will not be able to accept this version of the manuscript, but we would be willing to review again a much-revised version. We cannot, of course, promise publication at that time.

In particular, the discussion should be re-written to place results in context. Furthermore the work presented here needs more careful examination and additional experimentation that  explain why interaction of PR1 and TLP in the apoplast is important for wheat defence against a biotroph fungus that forms intimate connection with the host through formation of haustoria inside the cell.

If you decide to revise the manuscript for further consideration at PLOS Genetics, please aim to resubmit within the next 60 days, unless it will take extra time to address the concerns of the reviewers, in which case we would appreciate an expected resubmission date by email to plosgenetics@plos.org.

[LINK]

We are sorry that we cannot be more positive about your manuscript at this stage. Please do not hesitate to contact us if you have any concerns or questions.

Yours sincerely,

Gitta Coaker, PhD

Associate Editor

PLOS Genetics

Gregory P. Copenhaver

Editor-in-Chief

PLOS Genetics

Reviewer's Responses to Questions

**Comments to the Authors:**

Reviewer #1: In the manuscript the authors attempt to show that TaTLP1 and TaPR1 interact with one another and contribute to resistance against Puccinia triticina. They perform pull-down experiments to identifyTaPR1, and the interaction between TaTLP1 and TaPR1 was confirmed by yeast two-hybrid techniques, bimolecular fluorescence complementation (BiFC), and co-immunoprecipitation. They also used TaTLP1-TaPR1 co-silenced plants along with single knock down plants to show that there is an increase in susceptibility. The experiments indicate that the two proteins interact and these proteins contribute to resistance. But the major concern is that the experiments do not conclusively prove that this interaction contributes to resistance. The data can be interpreted as the proteins having overlapping functions or roles in resistance.

Some specific points:

The grammar and syntax throughout the manuscript need to be corrected. Some sentences are difficult to decipher.

Abstract:

The last line of the abstract can be re-written such that it reflects the findings about TaTLP1 and TaPR1 only instead of including all PR proteins in general.

Author Summary:

The first few lines of the summary can be shortened or deleted. The paragraph can begin with and discuss Thaumatins or TLP proteins and their role in general.

Line 38: The words “Defense responses” may be more suitable than “disease-resistance responses”,

Introduction:

Abbreviations and names should be explained at first. E.g. line 146: PHNT, the term Thatcher should also be explained. These terms and other abbreviated terms should be clearly described in the materials and methods section also.

Results:

Line 151-153: The fact that their expression profiles are similar under compatible and incompatible interaction does not show that they interact with each other, at this point. The sentence should be changed. The Y2H, BiFC co-localization experiments etc. subsequently described prove they do interact. So this fact can be claimed at a later point.

Figure 1: The gel photo is not distinct enough to see the indicated protein band(s). Moreover extra protein bands are visible in the -TaTLP1 at ca 30kDa, 25 kDa etc. Similarly extra protein bands are visible in the +TaTLP1 ca 45kDa, 50kDa etc.

Line 236: The experiments prove that these two proteins interact and that that the 52 amino acid conserved region of the TaPR1 is sufficient to bind the TaTLP1 but these do not indicate that this is “sufficient for defense to leaf rust”

Moreover the 52 amino acid truncated protein having only the conserved region may not bond with the same strength as that required for binding in vivo under diseased condition. So perhaps an experiment to show the strength of bonding between these two proteins (truncated TaPR1 and TaTLP1) in vitro or in transient expression experiments can shed some light. A preliminary prediction of Protein-Protein Interaction strength can be made using software.

Under diseased condition where there are several proteins which are differentially expressed, the situation can alter the affinity of the binding between these two proteins. This fact can be discussed in the discussion part.

Line 283-285: “antifungal activity of both TaTLP1-TaPR1 and TaTLP1 or TaPR1 alone significantly increased compared with control” So the inference is that these protein alone or in combination have antifungal activity. But the heading (line 269-270) does not agree with this. The heading seems to say that the protein in combination has more antifungal activity than the individual proteins. If that is so, then the sentence “that TaTLP1-TaPR1 could protect wheat by...” should also be changed to “the proteins alone or in combination can protect wheat by.....”.

Discussion

Much of the discussion is repetition of the results (e.g. Lines: 403-407, 427-460). Discussion should focus on the interpretation of the data with respect to prior findings and the significance of the findings.

Material and methods:

This section does not clearly state the host varieties used and the pathogen sources etc. The “materials” or organisms used and their sources should be mentioned.

References that can be included:

Anand A1, Zhou T, Trick HN, Gill BS, Bockus WW, Muthukrishnan S. Greenhouse and field testing of transgenic wheat plants stably expressing genes for thaumatin-like protein, chitinase and glucanase against Fusarium graminearum. J Exp Bot. 2003, 54(384):1101-11.

Chowdhury S, Basu A, Kundu S. Overexpression of a New Osmotin-Like Protein Gene (SindOLP) Confers Tolerance against Biotic and Abiotic Stresses in Sesame. Front Plant Sci. 2017;8:410. Published 2017 Mar 28. doi:10.3389/fpls.2017.00410

de Jesus-Pires C, Ferreira-Neto JRC, Pacifico Bezerra-Neto J, Kido EA, de Oliveira Silva RL, Pandolfi V, Wanderley-Nogueira AC, Binneck E, da Costa AF, Pio-Ribeiro G, Pereira-Andrade G, Sittolin IM, Freire-Filho F, Benko-Iseppon AM. Plant Thaumatin-like Proteins: Function, Evolution and Biotechnological Applications. Curr Protein Pept Sci. 2019 Mar 18. doi: 10.2174/1389203720666190318164905.

Faillace GR, Turchetto-Zolet AC, Guzman FL, de Oliveira-Busatto LA, Bodanese-Zanettini MH.Genome-wide analysis and evolution of plant thaumatin-like proteins: a focus on the origin and diversification of osmotins. Mol Genet Genomics. 2019 Apr 27. doi: 10.1007/s00438-019-01554-y.

Mackintosh CA1, Lewis J, Radmer LE, Shin S, Heinen SJ, Smith LA, Wyckoff MN, Dill-Macky R, Evans CK, Kravchenko S, Baldridge GD, Zeyen RJ, Muehlbauer GJ. Overexpression of defense response genes in transgenic wheat enhances resistance to Fusarium head blight. Plant Cell Rep. 2007 Apr;26(4):479-88. Epub 2006 Nov 11.

Van der Maelen E, Rezaei MN, Struyf N, Proost P, Verstrepen KJ, Courtin CM. Identification of a Wheat Thaumatin-like Protein That Inhibits Saccharomyces cerevisiae.

J Agric Food Chem. 2019 Sep 18;67(37):10423-10431. doi: 10.1021/acs.jafc.9b03432. Epub 2019 Sep 5.

Xing L.-P., Wang H.-Z., Jiang Z.-N., Ni J.-L., Cao A.-Z., Yu L., Chen P.-D.

Transformation of Wheat Thaumatin-Like Protein Gene and Analysis of Reactions to Powdery Mildew and Fusarium Head Blight in Transgenic Plants

(2008) Acta Agronomica Sinica, 34 (3) , pp. 349-354.

Reviewer #2: The manuscript by Wang et al., describes the importance of wheat TLP and PR1 in defence against wheat leaf rust. Authors discovered PR1 interaction with TLP. They investigated the function of these PR proteins by VIGS and also reported that PR1stabilize TLP protein.

the finding by wang et al that PR1 interact with TLP and their role in defence against leaf rust , is certainly interesting. However additional experiments will be needed to support some of the claims made by the authors. Also the manuscript needs significant edits to improve the english writing and the content.

1- There were several strong candidates in addition to PR1 as potential interactions of TLP . Some of these proteins have reported function in plant immunity and they had higher count than PR1 (Table1 1). Authors should examine interaction of these targets by Y2H and Co-IP?

2- Lines 195-196 and Fig 4B. Apoplast localization of PR1-TLP needs to be verified by co- localization of and apoplast marker.

3- Lines 216-234 and Fig 5B. Authors claim that TaPR1 stabilizes the accumulation of TaTLP protein. However the amount of protein in lane 2 is visibly less than lane 1 in fig 5 B. it is not clear why authors used MG132 when there is no evidence that PR1 and TLP are targeted to the proteasome pathway. Also myc control was used by there is no Myc antibody labelling. the order of lower panel Ha, GFP labelling should be revered so it correspond to the same order of the upper panel proteins .

3- L296-287: ViGS of PR1 and TLP1

A- why level of expression for PR1 and TLP1 are higher at 48 hrs compare to 24 hrs and goes down again at later time point. it is expected that as time progress level of expression of target genes should reduce or at least stay the same since the virus multiply and spreads. To knock down the expression of both PR1 and TLP1 a VIGS construct with fragments for PR1 and TLP1 should be used instead of mixing the two.

B- Fig. 8 needs proper labelling and caption need proper description to show which wheat cv was used. HR in the positive control leaf is not visible in the image. Pathogen growth seems to be the same in the PR1/TLP1 combination compared to PR1 or TLP1 alone as also seen in Fig 9.

C- Fig 10: H2O2 level needs to be quantified. it is hard to just from the image conclude that H2O2 level is reduced. Authors need to describe why they used SOD and CAT and how the change in tier expression correlate with the level of H2O2.

Finally manuscript needs to be edited extensively to improve the writing. Introduction should include a background about the leaf rust, genetic of resistance and what is known about the pathogen interaction with wheat.

results are not descriptive enough in some places and use of abbreviation should be described when t first appear in the text.

Discussion is mainly repeating the results and it does not discuss some of the main conclusion (e.g. stability of TLP1)

Reviewer #3: Line 23: In vitro, the…

Line 65: expand BcIEB1 and B. cinereal

Line 67: expand PevD1

Line 79: expand SsCP1

Line 80: expand GST

Page 4; Instead of talking about PTI and ETI please explain about race specific and APR resistance in wheat rust resistance. Also provide details that TLP1 and PR1 are found to be important in both these types (ref some papers such as Sr21, Sr13, Lr34 etc papers)

Also point that for the first time we report the interaction of these two molecules in wheat if its not been reported earlier

Line 146: Pt race PHNT

Page 10: Please indicate CAPE1 what is the amino acid region

Page 13: Please indicate that silencing of both TLP and PR produce slightly larger pustules as compared to the control

Line 408: CAPE1 is not required

Page 26: Please indicate the contribution of Xiong Du and Daqun Liu

Reviewer #4: In this work, the authors studyed the functions of TaLTP and TaPR1 ,and the interection between TaLTP and TaPR1 .Generally speaking, the work is well done , so the results are expectable.

According to the MS, I have some suggestions as following:

1.The English language is required to be improved by any reputable English language editing service. Some mistakes should be corrected through your manuscript such as TaTLP1 and TaPR1(TaTLP1 and TaPR1) from line 321 of page 12.

2.The descriptions of "Backgroud" section and "Discussion" are inadequate without the latest research progress, especially since 2018.

3.The title should be clearer,please see line 235 of page 9.

4. Please unify the statement:Puccinia triticina,leaf rust,and leaf rust fungus.

5.The line 282 of page 11:Microscopic observation showed that the germination of urediniospores and the growth of hyphae were significantly restricted with pure proteins,From the Fig7, it is not a hyphae but a germtube.

6.Please mark the bar of fig9 and fig10.

7.See line 369 of page 14:The areas of cell death and H2O2 accumulation was significantly lower at 120 hpi in TaTLP1/TaPR1-knockdown plants than that observed in BSMV:γ, TaTLP1 or TaPR1 alone inoculated leaves, please give the statistical result of the areas of cell death. Analyze the reasons in detail.

**Have all data underlying the figures and results presented in the manuscript been provided?**

Reviewer #1: Yes

Reviewer #2: Yes

Reviewer #3: Yes

Reviewer #4: Yes

PLOS authors have the option to publish the peer review history of their article (what does this mean?). If published, this will include your full peer review and any attached files.

Reviewer #1: Yes: Dr. Surekha Kundu

Molecular and Applied Mycology and Plant Pathology Laboratory

Department of Botany

University of Calcutta, India

Reviewer #2: No

Reviewer #3: No

Reviewer #4: No

---

## [Decision Letter · Decision Letter 1]

14 Feb 2020

Dear Dr Wang,

Thank you very much for submitting your Research Article entitled 'TaTLP1 interacts with TaPR1 to contribute to wheat defense responses to leaf rust fungus' to PLOS Genetics. Your manuscript was fully evaluated at the editorial level and by independent peer reviewers. The reviewers appreciated the attention to an important topic but identified some aspects of the manuscript that should be improved.

Please address English Language writing in the paper and edit for clarity. Also, there are not sufficient details provided for the statistical analyses. What do the error bars represent? SE, SD? Not all the figures indicate the statistical test used. If ANOVA was used, what mean separation statistics were used after a significant F test? what N represents should also be included (individual plants, individual leaves, etc).

We therefore ask you to modify the manuscript according to the review recommendations before we can consider your manuscript for acceptance. Your revisions should address the specific points made by each reviewer.

[LINK]

Yours sincerely,

Gitta Coaker, PhD

Associate Editor

PLOS Genetics

Gregory Copenhaver

Editor-in-Chief

PLOS Genetics

Reviewer's Responses to Questions

**Comments to the Authors:**

Reviewer #1: The data are now better presented. The authors made the recommended corrections. There remain many syntax errors. The manuscript still needs to be rewritten in lucid English, starting from the abstract.

Reviewer #3: No comments

Reviewer #4: Evidence from in vivo experiments is not sufficient

**Have all data underlying the figures and results presented in the manuscript been provided?**

Reviewer #1: Yes

Reviewer #3: Yes

Reviewer #4: Yes

PLOS authors have the option to publish the peer review history of their article (what does this mean?). If published, this will include your full peer review and any attached files.

Reviewer #1: No

Reviewer #3: No

Reviewer #4: No

---

## [Editor Report · Decision Letter 2]

11 Mar 2020

Dear Dr Wang,

We are pleased to inform you that your manuscript entitled "TaTLP1 interacts with TaPR1 to contribute to wheat defense responses to leaf rust fungus" has been editorially accepted for publication in PLOS Genetics. Congratulations!

Yours sincerely,

Gitta Coaker, PhD

Associate Editor

PLOS Genetics

Gregory P. Copenhaver

Editor-in-Chief

PLOS Genetics

Comments from the reviewers (if applicable):

**Data Deposition**

http://datadryad.org/submit?journalID=pgenetics&manu=PGENETICS-D-19-01570R2

**Press Queries**

---

## [Editor Report · Acceptance letter]

5 Jun 2020

PGENETICS-D-19-01570R2 

TaTLP1 interacts with TaPR1 to contribute to wheat defense responses to leaf rust fungus 

Dear Dr Wang, 

We are pleased to inform you that your manuscript entitled "TaTLP1 interacts with TaPR1 to contribute to wheat defense responses to leaf rust fungus" has been formally accepted for publication in PLOS Genetics! Your manuscript is now with our production department and you will be notified of the publication date in due course.

With kind regards,

Kaitlin Butler

PLOS Genetics

On behalf of:
